# A Strategy for Preparing Solid Polymer Electrolytes Containing In Situ Synthesized ZnO Nanoparticles with Excellent Electrochemical Performance

**DOI:** 10.3390/nano12152680

**Published:** 2022-08-04

**Authors:** Yinsi Xu, Jun Li, Wanggen Li

**Affiliations:** School of Materials Engineering, Shanghai University of Engineering Science, Shanghai 201620, China

**Keywords:** in situ synthesized ZnO, crystallinity, coin cells, ion conductivity, cycle stability

## Abstract

ZnO nanoparticles were successfully in situ synthesized in the form of PEO–COO^−^ modified ZnO by a three-step method, based on which the solid polymer electrolytes (SPEs), based on polyethylene oxide (PEO) with excellent electrochemical performance, were prepared. The evolution of the electrochemical and mechanical performances of the SPEs with the ZnO content (0–5 wt.%) was investigated in detail. The mechanical property of the SPEs demonstrated a Λ-shaped change trend as increasing the ZnO content, so that the highest value was acquired at 3 wt.% ZnO. The SPE containing 3 wt.% ZnO had the most outstanding electrochemical performance, which was significantly better than that containing directly-added ZnO (2 wt.%). Compared with the latter, the ion conductivity of the former was improved by approximately 299.05% (1.255 × 10^−3^ S·cm^−1^ at 60 °C). The lithium-ion migration number was improved from 0.768 to 0.858. The electrochemical window was enhanced from 5.25 V to 5.50 V. When the coin cell was subject to the cycling (three cycles in turn from 0.1 C to 3 C, and subsequent fifty cycles at 1 C), the 68.73% specific capacity was retained (106.8 mAh·g^−1^). This investigation provides a feasible approach to prepare the SPEs with excellent service performance.

## 1. Introduction

To date, lithium-ion batteries (LIBs) have been widely used in the fields of energy storage, clean transportation vehicles, and aerospace [1,2,3]. The widely available organic liquid electrolyte in commercial LIBs has become a major safety concern due to its leakage, volatile, decomposition, combustible, and explosive characteristics. Moreover, lithium dendrites can grow easily in the liquid electrolyte, connecting the two electrodes and causing the short circuit of LIBs, which not only leads to security risks, but also greatly shortens the service life [4,5]. With respect to the above-mentioned issues, all-solid-state LIBs are increasingly becoming the focus of international attention, in which the liquid electrolyte is displaced by a safe and reliable solid electrolyte [6,7,8].

As an alternative solid electrolyte, the solid polymer electrolyte (SPE) has elicited extensive concern due to its good film-forming property, flexibility, safety, and interface compatibility [9]. The matrix used in the SPE can be divided into polyacrylonitrile [10,11], polyvinylidene fluoride (PVDF) [12,13], polymethyl methacrylate [14,15], and polyethylene oxide (PEO) [16,17,18]. Among them, PEO is a polymer of greatest interest in the SPE because of its low cost, excellent mechanical performance, high dielectric constant, and good compatibility with lithium salts and lithium metals [3,16,18]. Many studies have indicated that Li^+^ mainly migrates in the amorphized zones of PEO [3,5,16,17,18]. To better promote ion transportation and obtain good ion conductivity, some methods such as blending polymers and adding plasticizers and nanofillers have been applied to reduce the crystallinity of PEO [19,20]. Blending PVDF and PEO have proved to be an effective alternative [8], based on which Al_2_O_3_ [21], SiO_2_ [22], TiO_2_ [23], and other inorganic fillers were further introduced into the SPEs to improve mechanical and electrochemical performance.

The used inorganic fillers usually present a good stiffness, thermal stability, and chemical stability, which can be combined with the good toughness, dielectric property, and processing ability involved in the polymers [12,13,14,15,16,17,18,21,22,23,24,25,26]. Consequently, the composites composed of the inorganic fillers and the polymers demonstrate an excellent performance. As an amphoteric oxide, nano ZnO is expected to play a role similar to the above-mentioned inorganic fillers, which has been confirmed in some related reports [24,25]. However, there have been few reports about preparing the SPEs composed of ZnO, PEO, PVDF, and LiClO_4_ up to now. Our research group first carried out the related research and confirmed that directly adding a certain amount of ZnO (2 wt.%) not only improved the electrochemical performance with respect to ion conductivity (1.426 × 10^−4^ S·cm^−1^ for 0 wt.% ZnO and 3.145 × 10^−4^ S·cm^−1^ for 2 wt.% ZnO), lithium ion migration number (0.645 for 0 wt.% ZnO and 0.768 for 2 wt.% ZnO), electrochemical window (5.0 V for 0 wt.% ZnO and 5.25 V for 2 wt.% ZnO), excellent cycle stability (549 h for 0 wt.% ZnO and more than 1000 h for 2 wt.% ZnO), and rating capability (short circuit for 0 wt.% ZnO and 63.02% of capacity retained for 2 wt.% ZnO when the current density was improved from 0.1 C to 3 C and then recovered to 1 C) at 60 °C, but also enhanced the mechanical performance in terms of tensile strength, breaking strength, and elongation at break (1.281 MPa/2.349 MPa, 0.766 MPa/1.401 MPa, and 66.167%/241.911% for 0 wt.% ZnO and 2 wt.% ZnO, respectively) [26]. However, ZnO is directly added into the SPEs, which may cause some adverse consequences such as easy agglomeration and contamination, weak compatibility, and poor strength between ZnO particles and polymers. As a result, the improvement in electrochemical and mechanical performance is greatly restricted. A method called in situ synthesis was recently proposed to introduce ceramic particles through reactions occurring among different elements or compounds [5,19,27,28,29,30,31,32,33,34]. The in situ synthesized ceramic particles possess clean surface and good dispersibility, excellent compatibility with the polymers, and high interface binding strength. This method is a new strategy for further promoting the mechanical and electrochemical performance of the SPEs, and its effectiveness has been proven by the related investigations. Ma et al. [27] in situ synthesized TiO_2_ in the polymer matrix (poly(ethylene glycol) methyl ether methacrylate/stearyl methacrylate) to prepare the SPE. The electrolyte demonstrated a large migration number of 0.37, a high ionic conductivity of 1.10 × 10^−4^ S·cm^−1^ (30 °C), and a wide electrochemical window of 5.44 V. Huang et al. [5] prepared the solid electrolyte by in situ synthesizing Li_6.4_La_3_Zr_1.4_Ta_0.6_O_12_ in the polymer matrix (polyethylene carbonate). Given the integrated interface and extended Li^+^ migration channel, the ion conductivity reached 7.8 × 10^−5^ S·cm^−1^ at room temperature. It also presented a large electrochemical window of 4.5 V and a high lithium-ion migration number of 0.556. Xu et al. [28] constructed a 3D network structure of SPE (PEO@SiO_2_) by an in situ self-assembling strategy. The crystallinity of PEO was greatly reduced, so the ionic conductivity reached 1.1 × 10^−4^ S·cm^−1^ at 30 °C. The good solid–solid interface stability also improved the electrochemical window (4.8 V at 90 °C). However, the related reports about in situ synthesized ZnO in the blending polymer (PEO+PVDF) are lacking. 

On the basis of our previous investigation about directly adding ZnO nanoparticles into the matrix of PEO+PVDF, we successfully prepared the PEO-in situ synthesized ZnO/PEO/PVDF/LiClO_4_ SPEs. The effect of in situ synthesized ZnO content on the microstructure and electrochemical/mechanical performance was investigated comprehensively. The action mechanism of ZnO was revealed. Moreover, the influence of microstructure on electrochemical/mechanical performance was studied. Finally, the suitable content of in situ synthesized ZnO was determined. 

## 2. Experimental Methods

### 2.1. Materials

Poly(ethylene glycol) (PEG, Mw = 4000, Greagent, Titan^®^, Shanghai, China), potassium permanganate (KMnO_4_, 99.5 wt.%, Greagent), potassium hydroxide (KOH, Mw = 56.11, Sinophary, Shanghai, China), zinc hydroxide (Zn(OH)_2_, Mw = 99.38, Damao Chemical Reagent Factory, Tianjin, China), lithium hydroxide (LiOH, 99.99 wt.%, Aladdin, Shanghai, China), sulfuric acid (H_2_SO_4_, 96 vol.%, Adamas, Shanghai, China), diethyl ether (C_4_H_10_O, 99.5 vol.%, Greagent), and ethanol (CH_3_CH_2_OH, 99.7 vol.%, Greagent) were used to in situ synthesize ZnO. In addition to the above reagents, the chemical reagents used to prepare the SPEs included *N*,*N*-dimethylformamide (DMF, 99.9 vol.%, Aladdin), PVDF (Mw = 900,000, Arkema, Shanghai, China), LiClO_4_ (99.99 wt.%, Aladdin), and PEO (Mw = 1,000,000, Aladdin). Besides the above-mentioned PVDF, three reagents, namely, lithium iron phosphate (LFP, Battery level, Taiyuan Lizhiyuan Battery Co., Ltd., Taiyuan, China), acetylene black (SP, Battery level, Taiyuan Lizhiyuan Battery Co., Ltd.), and *N*-methylpyrrolidone (NMP, 99.9 vol.%, Aladdin), were selected to prepare the cathode.

### 2.2. Preparation of In Situ ZnO and the SPEs

#### 2.2.1. Preparation of In Situ ZnO

ZnO was in situ prepared (Figure 1): hydrocarbyl was oxidized into carboxyl in PEG (Step 1), H in carboxyl of PEG was replaced by Zn (Step 2), and Zn was further transformed into ZnO (Step 3).

Step 1: About 2 g of dried PEG was dissolved into 20 mL deionized water by magnetic stirring for 30 min. The dried KOH and KMnO_4_ were weighed in a certain molar ratio (PEG:KOH:KMnO_4_ = 1:5:4) and dissolved into 10 and 30 mL deionized water, respectively. The PEG solution needed to be alkalized by introducing a few drops of KOH solution prior to mixing the three solutions, and the KOH and KMnO_4_ solutions were slowly dribbled into the above PEG solution. The hydrocarbyl was oxidized into the carboxyl in PEG through the reactions among PEG, KMnO_4_, and KOH, approximately lasting for 5–6 h under shading and magnetic stirring. The solution color was gradually transformed from initial amaranth (MnO_4_^−^) into green (MnO_4_^2−^ as the intermediate product), and finally into brown (MnO_2_). Insoluble MnO_2_ was filtered to obtain a clear solution. The solution was subsequently neutralized with diluted sulfuric acid (0.1 mol/L) prepared in advance (the pH measured by the pH meter was pH 7). PEO containing carboxyl group (PEO–COOH) was obtained by the above treatment.

Step 2: Approximately 0.3 g dried Zn(OH)_2_ was added to the above solution and reacted for 5 h under magnetic stirring at 75 °C, during which (PEO–COOH)_2_ was replaced by Zn(PEO–COO)_2_. Excess Zn(OH)_2_ was filtered from the above solution, dried, and weighed. The mole number of Zn(OH)_2_ participating in the reaction was calculated. The solution subjected to filtering was dried to remove excess water in an electric constant-temperature drying oven (DHG-9023A, Shanghai Jinghong Experimental Equipment Co., Ltd., Shanghai, China) at 65 °C for 12 h. The product was added into absolute ethanol (30 mL), fully stirred, and filtered to remove the insoluble substances (K_2_SO_4_). The solution after filtering was evaporated to remove absolute ethanol in the electric constant-temperature drying oven at 40 °C for 10 h and then dried to obtain the resultant product (Zn(PEO–COO)_2_) in a vacuum oven (DZF-6020, Shanghai Jinghong Experimental Equipment Co., Ltd., Shanghai, China) at 100 °C for 12 h. 

Step 3: Zn(PEO–COO)_2_ was weighed and dissolved into 25 mL absolute ethanol. According to the molar ratio of Zn to Li (1:1.4), anhydrous LiOH was weighed and dissolved into 15 mL absolute ethanol. After the above two substances were fully dissolved under magnetic stirring, the ethanol solution containing LiOH was slowly added to the ethanol solution containing Zn(PEO–COO)_2_ with a dropper. ZnO was in situ synthesized under specific magnetic stirring for 5 h. The obtained solution was evaporated to remove ethanol in the electric constant-temperature drying oven at 40 °C for about 12 h after the above reaction was completed, and then transferred into a separatory funnel after adding it into 20 mL anhydrous ethyl ether. The liquid was fully shaken and divided into two layers after standing; the lower suspension was quickly allowed to flow out. The suspension was dried to eliminate the residual ethanol and ether in the vacuum oven at 100 °C overnight. The final product was PEO–COO^−^-modified ZnO and named PEO-in situ ZnO.

#### 2.2.2. Preparation of the SPEs

To prepare SPEs with different in situ ZnO contents, the mass of pure PEO, PVDF, and PEO-in situ ZnO based on the required ZnO content (from 0 wt.% to 5 wt.%) was first weighed. The mass ratio of PEO in the mixture (pure PEO and PEO-in situ ZnO) and PVDF was 7:3. The above pure PEO and PVDF were added to 25 mL DMF solvent to form a uniform sol under magnetic stirring for 4 h at 80 °C. The temperature was adjusted to 70 °C. PEO-in situ ZnO was added into the above sol and mixed for 2 h under magnetic stirring. Finally, LiClO_4_ was introduced to the above product based on the molar ratio of 10:1 (EO (in the mixture of pure PEO and PEO-in situ ZnO):Li) under magnetic stirring at 70 °C for 6 h. The SPE film was prepared on an automatic thick film coater (MSK-AFA-III, Hefei Kejing Material Technology Co., Ltd., Hefei, China). First, the prepared electrolyte sol was poured on the polytetrafluoroethylene (PTFE) plate (300 × 200 × 10 mm^3^), and the sol film was formed by moving the scraper at a constant speed (2 mm·s^−1^). An interval of 2 mm between the scraper and the plate was maintained to ensure the film’s homogeneity. After drying the PTFE plate coated with sol film in the oven at 70 °C for 2 h, it was transferred into the vacuum oven at 55 °C for 24 h to remove the remaining solvent. After cooling to below 30 °C, the SPE film was removed from the PTFE plate, and the circular film with a diameter of 19 mm was cut by using an undefiled handy microtome (T07, Hefei Kejing Material Technology Co., Ltd., Hefei, China). 

### 2.3. Preparation of the SPEs with Directly-Added ZnO

In order to compare the electrochemical performance of the SPEs with in situ ZnO and directly-added ZnO, the SPEs with different contents of directly-added ZnO (1 wt.%, 2 wt.%, 3 wt.%, 4 wt.%, and 5 wt.%) were also prepared by the blending method. Firstly, a certain amount of commercial nano-ZnO (99 wt.%, particle size < 100 nm, Aladdin) was directly added into 25 mL DMF, and uniformly dispersed under the action of magnetic stirring and ultrasonic treatment. Secondly, PEO and PVDF (a mass ratio of 7:3) were added into the above solution, and magnetically stirred at 80 °C for 4 h. Finally, LiClO_4_ (the molar ratio of EO (calculated as PEO):Li was 10:1) was added into the above product, and magnetic stirring was performed at 70 °C for 8 h to make LiClO_4_ uniformly dispersed in the above product. The film forming process is the same as Section 2.2.2. Based on our previous study (seeing Ref. [26]), the SPE with 2 wt.% directly-added ZnO demonstrated the best electrochemical performance, which was employed for comparison. 

### 2.4. Preparation of the Cathode

Adhesive PVDF (40 mg) was added to 1040 mg of NMP solvent, which was stirred with a glass rod until PVDF was completely dissolved. LFP (1900 mg) and SP (Super-P) (60 mg) were placed into an agate mortar and fully ground for 1 h. SP belongs to a kind of conductive carbon black consisting of many small particles, in which primary aggregates with 150~200 nm size are distributed around the active substance and cause the formation of a branched conductive network. The introduction of SP aims to reduce the battery resistance and improve the ion conductivity. The mixture was added into the above solution, which was stirred for 1 h with a glass rod to form a uniform slurry. The prepared slurry with a thickness of 100 μm was coated on aluminum foil by an automatic thick film coater. The coated aluminum was placed in the vacuum oven at 60 °C for 36 h to remove the remaining NMP. Finally, the resultant coated aluminum as the cathode was cut into a circular piece with a diameter of 14 mm by using an undefiled handy microtome.

### 2.5. Material Characterization

An X-ray diffractometer (XRD, D2-PHASER, Bruker, Karlsruhe, Germany) using Cu Kα radiation and a Fourier infrared spectrometer (FT-IR, IS-10 IR, Nicolet, Shanghai, China) with a scope of 400–4000 cm^−1^ were used to analyze the phase structure and functional groups of the SPEs, respectively. The crystallinity of the SPEs was quantitatively characterized by differential scanning calorimetry (DSC, Q2000 V24.11 Build 124 NAICHI, Shanghai, China) in nitrogen with a certain temperature range (−30 °C–120 °C). A field emission scanning electron microscope (FE-SEM, S-4800, HITACHI, Tokyo, Japan) was employed to observe the morphological characteristics of the SPEs. A universal testing machine (LDW-5, Shanghai Songdun Instrument Manufacturing Co., Ltd., Shanghai, China) was used to test the mechanical properties of the SPEs with a size of 5 × 1 cm^2^ at a tensile rate of 5 mm·min^−1^.

### 2.6. Electrochemical Tests

Before testing, the SPEs were assembled into coin cells (CR2032) in a glovebox filled with argon (Super 1220/750/900, Shanghai Mikrouna Electromechanical Technology Co., Ltd., Shanghai, China). In this work, the following five tests were carried out. Electrochemical impedance spectroscopy (EIS) tests were executed on the coin cells (SS (stainless-steel sheet)/SPE/SS), which were used to ensure the conductivity of the SPEs. The electrochemical window was obtained by linear sweep voltammetry (LSV) tests on the coin cells (Li (lithium sheet)/SPE/SS). The EIS and *i*-t polarization tests executed on the coin cells (Li/SPE/Li) were used to calculate the lithium-ion migration number ($t_{\mathrm{ion}}$). All the above tests were performed by using an electrochemical workstation (CHI 760E, CH Instruments Inc., Shanghai, China). Moreover, the coin cells (Li/SPE/Li) were used for the charge/discharge cycle stability tests on a Neware battery tester (CT4008, NEWARE, Shenzhen, China). The influence of the SPEs on the rate performance of the coin cells was also verified. The coin cells (LFP/SPE/Li) were subjected to constant current charge/discharge tests with a voltage scope of 2.7–4.2 V by using a Neware battery tester.

SEM was employed to investigate the morphologies of lithium dendrites formed on the surface of the lithium sheets in the coin cells subject to long-term charge/discharge tests. Prior to the observation, the coin cells were disassembled by the cleaned tweezers, pliers, and other tools in the glove box filled with argon. In the glove box, the edges of the coin cells were firstly pried using the tweezers and the cathode shells of the coin cells were removed. The components were taken out and the lithium anodes were stripped from the components using the tweezers. During the stripping, it can be observed that the anode was easily separated from the SPE without ZnO due to the SPE preserved completely. On the contrary, it was difficult to separate the anode from the SPE with 3 wt.% ZnO, leaving the incomplete fragments on the anode. This indicated that the adhesion between the SPE and the anode can be improved by introducing ZnO into the SPE. 

## 3. Results and Discussion

### 3.1. Structural Characterization

The phase composition change of the SPEs and three pure substances (PEO, PVDF, and ZnO) were studied by XRD (Figure 1a). With respect to pure PEO, it demonstrated high crystallinity due to two intense and pointed diffraction peaks at 23.2° and 19.3°. For pure PVDF, two adjacent peaks were identified at 18.3° and 19.8°, which originated from the existence of the α phase [22,35]. Regarding pure ZnO, the d values of all peaks were in accordance with those of ZnO recorded in JCPDS (NO. 01-076-0704). When LiClO_4_ was added into the mixture of PEO and PVDF (SPE with 0 wt.% ZnO), the two main peaks associated with PEO were still observed, but their intensity was greatly reduced, implying that crystallinity further decreased (Figure 1a). Moreover, the peak (18.3°) related to the α phase in PVDF completely disappeared, and the other peak (19.8°) shifted 0.5° to the right, indicating that the α phase with high crystallinity was transformed into the β phase with low crystallinity [22,35]. These results indicated that the crystallinity of the SPE could be reduced to a certain extent by mixing PEO, PVDF, and LiClO_4_, which originated from the interaction among the three. For PEO and PVDF, the interaction between the ether “O” atom in PEO and the “F” atom in PVDF destroyed the order of the polymer chains, causing the crystallinity of the copolymer phase to decrease [36,37]. When LiClO_4_ was also mixed, Li^+^ demonstrated strong complexation ability to the C–F bond in PVDF and the ether oxygen bond in PEO. Moreover, the chain structure of PEO and PVDF was highly likely to be cross-linked through Li^+^ due to its strong multiple coordination ability. The above two effects promoted the further reduction in crystallinity of the copolymer. Sengwa et al. [35] prepared the SPE with the PVDF/PEO blend as the matrix and LiClO_4_ as the ion dopant. Their results showed that the crystallinity of the 75 wt.% PVDF/25 wt.% PEO film (without salt) was 34.7%. After adding 5 wt.% LiClO_4_, the crystallinity was drastically dropped to 22.5%. This proved the positive role in reducing the crystallinity by introducing the lithium salt.

When different contents of ZnO were in situ synthesized in the SPEs (1–5 wt.%), three obvious changes were observed (Figure 1a). First, the two strongest peaks of ZnO appeared at 31.7° and 36.2° and presented an upward intensity with the increase in ZnO content. Moreover, the two strongest peaks related to PEO were first decreased and then increased in strength, leading to their weakest intensity obtained in the SPE with 3 wt.% ZnO. Finally, the peak of PVDF in the SPEs containing ZnO was low and wide compared with that without ZnO. The above-mentioned changes demonstrated that introducing a suitable content of ZnO had a significant effect on reducing the crystallinity of the SPEs. This change was closely associated with the interaction between ZnO and PEO. The large electrostatic reciprocity between the ether oxygen bonds in the PEO chain and the zinc atoms in ZnO could effectively destroy the PEO spherulites to a certain extent. The uniformly dispersed ZnO fillers served as the cross-linking points to promote random cross-links among the polymer molecules. Przyluski et al. [38] applied the effective medium theory to explore the reciprocity between the adulterant of Al_2_O_3_ and the polymer in the PEO–NaI–Al_2_O_3_ system. When Al_2_O_3_ was introduced to the polymer, the reactions occurred at the interface of Al_2_O_3_ and PEO, promoting the formation of an amorphous polymer phase. Thus, the amorphous region in the electrolyte was enlarged. Wang et al. [39] used the large-scale molecular dynamics simulation method to study the interaction in the nanoparticle/polymer system. Their results showed a certain adsorption effect between nanoparticles and polymer molecules. By introducing more nanoparticles without agglomeration, the intervals among the dispersed nanoparticles would be greatly reduced because the polymer molecules in the specific zone would be adsorbed by multiple nanoparticles around them. The generated adsorption force at different directions would seriously destroy the orderly arrangement of those polymer molecules; namely, the crystallinity would be reduced. However, excessive introduction (higher than 3 wt.%) produced the opposite effect, which may be due to the efficient reciprocity between ZnO and PEO, which was greatly weakened as a result of the agglomeration of ZnO particles. This change had also been reported in similar studies. Hou et al. [29] studied the crystallinity of hyper-branched poly(amine-ester) (HBPAE)/HBPAE-g-TiO_2_/PEO electrolyte by in situ synthesis of hyperbranched poly(urethane)-doped TiO_2_/PEO composite materials. The lowest crystallinity of the SPEs was acquired when the TiO_2_ content reached 15 wt.%, while the value showed a slight improvement as the TiO_2_ content further increased.

The crystallinity of the SPEs was quantitatively analyzed through DSC tests (Figure 1b). An endothermic peak was observed on each corresponding curve obtained at every SPE, at which the corresponding peak temperature was denoted as the melting temperature ($T_{m}$), and the surrounding area signified the melting enthalpy ($\Delta H_{m}$). The following formula can accurately calculate the crystallinity of the SPEs [40]:(1)$$X_{c}=\left(\frac{\Delta H_{m}}{\Delta H_{m}^{0}}\right)\;\times \;100\%$$
where $X_{c}$ represents the crystallinity, and $\Delta H_{m}^{0}$ represents the standard melting enthalpy of pure PEO ($\Delta H_{m}^{0}$ = 213.7 J·g^−1^). 

As shown in Table 1, pure PEO and PVDF demonstrated the high crystallinity (80% for PEO and 50–60% for PVDF). When LiClO_4_ was introduced into the mixture of the two polymers (PEO and PVDF) (SPE with 0 wt.% ZnO), the crystallinity of the SPE was drastically reduced to 7.11%. Moreover, the content of ZnO significantly influenced the crystallinity of the SPEs, namely, the crystallinity was decreased by increasing the ZnO content from 0 wt.% to 3 wt.% (7.11%, 3.72%, 3.20%, and 2.04% for 0 wt.%, 1 wt.%, 2 wt.%, and 3 wt.%, respectively) and then increased with the ZnO content (4.60% and 4.89% for 4 wt.% and 5 wt.%, respectively). In general, the crystallinity in the SPEs with ZnO was lower than that without ZnO, indicating that the introduction of ZnO contributed to the reduction in crystallinity of the SPEs. However, this result did not mean that the low crystallinity could be obtained by introducing more ZnO. In this study, the suitable content of ZnO was 3 wt.% due to the lowest crystallinity obtained, at which the crystallinity was sharply reduced by approximately 71.31%. The excess introduction (4 wt.% and 5 wt.%) of ZnO played an opposite effect, which was due to the agglomeration of ZnO particles. This speculation was confirmed by subsequent SEM observations. 

The FTIR spectra of the SPEs and three pure substances (PEO, PVDF, and ZnO) were measured to investigate the structural changes in the SPEs with the ZnO content (Figure 1c). The characteristic peaks of pure PEO and PVDF and their corresponding band assignments were found in References [26,41]. For pure ZnO, a characteristic peak related to Zn-O was observed at 433 cm^−1^ [24], and two other peaks also appeared at 1631 and 3444 cm^−1^. The two peaks may be associated with the hydroxyl groups formed between ZnO and the adsorbed water on its surface. When three substances (PEO, PVDF, and LiClO_4_) were mixed, the four absorption peaks of PEO at 962, 1414, 1981, and 2918 cm^−1^ shifted to 949, 1410, 1965, and 2883 cm^−1^ toward the right, respectively. Moreover, two absorption peaks at 1097 and 1059 cm^−1^ were replaced by a broad peak at 1072 cm^−1^. Regarding PVDF, two changes were observed: three absorption peaks at 833, 1230, and 1167 cm^−1^ disappeared, and the three other absorption peaks shifted from 874, 1070, and 1402 cm^−1^ to 877, 1072, and 1410 cm^−1^, respectively. These phenomena were mainly attributed to the interaction among the three. Moreover, a new peak appeared at 623 cm^−1^, which confirmed the existence of ClO_4_^−^ [28]. 

When ZnO was in situ synthesized in the SPEs (Figure 1c), the F–C–F asymmetrical stretching peak in PVDF moved from 877 cm^−1^ to 879 cm^−1^ (1–5 wt.%). The C–O–C asymmetric stretching of PEO at 1072 cm^−1^ in the SPE without ZnO first moved to 1091 (1 wt.%), 1088 (2 wt.%), and 1086 cm^−1^ (3 wt.%) toward the left and then shifted to 1095 (4 wt.%) and 1093 cm^−1^ (5 wt.%) toward the opposite direction with increasing ZnO content. A weak peak appeared at 1060 cm^−1^ when introducing ZnO, and its intensity presented an upward trend with the increase in ZnO content. The above changes proved that the introduced ZnO interacted with PEO and PVDF. The dissociation of the aforementioned lithium salt was quantitatively characterized by the peak area at 623 cm^−1^ (Figure 1d), which was increased from 3.28 (0 wt.%), 3.56 (1 wt.%), and 3.70 (2 wt.%) to 3.81 (3 wt.%) and then decreased to 3.49 (4 wt.%) and 3.46 (5 wt.%) (Figure 1d). These results showed that Li^+^ could be sufficiently released when 3 wt.% ZnO was introduced, which was beneficial to the improvement in lithium-ion migration number and ionic conductivity [40]. 

Figure 2 showed the SEM images of the SPEs with different ZnO contents. For the SPE without ZnO (Figure 2(a1,b1)), the surface was rough due to a considerable number of irregular crystals involved in the SPE. When 1 wt.% (Figure 2(a2,b2)) and 3 wt.% ZnO (Figure 2(a3,b3)) were in situ synthesized in the SPEs, the crystals vanished and their surfaces were no longer rough, which indicated that the crystallinity of the SPEs was reduced. Some fine white clusters could be identified in the two SPEs. The volume fraction of clusters in the SPE containing 3 wt.% ZnO was higher than that of clusters in the SPE containing 1 wt.% ZnO. When the ZnO content reached 5 wt.% (Figure 2(a4,b4)), some clusters agglomerated to form coarse particles (see the local magnified image in Figure 2(a4)), around which many fine cracks initiated and propagated due to the mismatch in elastic modulus between ZnO and copolymer when they shrank during the volatilization of the solvent (DMF) (Figure 2(b4)). The agglomeration resulting from the excess introduction of ZnO not only weakened the interaction between ZnO and copolymer, but also destroyed the whole continuity of the copolymer. This change could greatly reduce the mechanical properties of the SPEs and deteriorate their electrochemical performance. Mapping scanning was performed to distinguish the compositions of the clusters in the SPE with 5 wt.% ZnO. As shown in Figure 2(c1–c4), some bright clusters can be clearly identified in the SEM image (Figure 2(c1)), which were rich in Zn and O but poor in C. Combined with the XRD results, it can be concluded that ZnO particles (bright clusters) were in situ synthesized in the polymer matrix. There were no clear and regular interfaces between the bright clusters (rich in ZnO) and the other zones, indicating that ZnO was successfully integrated into the polymer matrix. The bright clusters should include a large number of fine ZnO particles. Based on the information in Figure 2(b3,c1), the average size of the bright clusters was approximately 40 nm and 500 nm, respectively. It can be inferred that the nanometer-sized ZnO particles were in situ synthesized in the SPE with 3 wt.% ZnO. By comparing Figure 2(b3) (3 wt.% ZnO) and Figure 2(c1) (5 wt.% ZnO), it can be found that the bright clusters with similar size were uniformly distributed in the whole polymer matrix (SPE with 3 wt.% ZnO). However, the bright clusters with significant differences in size (60–1000 nm) were observed to be scattered in the polymer matrix (SPE with 5 wt.% ZnO). This also provided the support for the agglomeration of ZnO particles when the content of ZnO was above 3 wt.%.

### 3.2. Mechanical Properties

For all solid-state lithium batteries, the mechanical properties of the SPEs were regarded as an essential index evaluating the practicability of lithium batteries. Long-time repetitive Li^+^ intercalation/deintercalation during charging and discharging will induce protuberance and sharp spikes (called lithium dendrites), which tend to pass through the electrolyte and cause short circuits and even explosion of the batteries. The safety of the lithium batteries should be the utmost concern during their practical service. The SPEs with excellent mechanical properties exhibited strong resistance to growth of lithium dendrites, greatly improving the service safety and prolonging the service life. In this study, the mechanical property was evaluated by unidirectional static tensile tests, and some mechanical indexes such as breaking strength (MPa), tensile strength (MPa), yield strength (MPa), and elongation at break (%) could be determined from the resultant stress–strain curves (Figure 3). The values of these mechanical indexes were shown in Table 2. The yield strength/tensile strength/breaking strength presented a gradual uptrend with the addition of ZnO content. The three values of the SPE with 3 wt.% ZnO were increased by approximately 142.94%, 270.73%, and 272.72% when compared with those of the SPE without ZnO. These observations indicated that the appropriate content of ZnO could enhance the resistance to plastic deformation and fracture. Numerous active sites on the surfaces of nanoparticles could interact with the polymer and form a strong coordinated covalent bond [39,42]. When the SPEs containing ZnO particles were subjected to external force, the polymer molecular segments moved along the external force direction, accompanied with comparative movement among the molecular chains. The strong interaction force between ZnO particles and polymer molecules produced great resistance to polymer molecular movement and crack initiation. Meanwhile, the high-hardness ZnO particles as part of the 3D network nodes inhibited the movement of polymer molecular chains and crack propagation. Consequently, the above-mentioned strengths were improved greatly. Besides the strength, the deformation ability was also significantly increased because the elongation at break (%) of the SPE with 3 wt.% ZnO was enhanced by approximately 568.91% compared with that without ZnO. This phenomenon was attributed to the strong interaction around ZnO particles promoting the movement of molecular chains occurring at different directions when undergoing the external force. Regarding the improvement in strength and plastic deformation, the surrounded area by the stress–strain curve was increased as the ZnO content increased from 0 wt.% to 3 wt.%; this result implied that the SPE with ZnO could absorb more energy prior to the fracture; namely, the fracture was improved [40].

However, when more ZnO was introduced into the SPEs (4 wt.% and 5 wt.%), the three strengths and elongation at break was not continuously raised but decreased. The values of the SPE with 5 wt.% were reduced by approximately 49.33%, 57.80%, 57.90%, and 89.50% when compared with those with 3 wt.%. The sharp reduction in these indexes was closely associated with the agglomeration of ZnO particles. Fine cracks around the coarse agglomerated ZnO particles could be regarded as the other essential factor for this change. With respect to the mechanical properties, the suitable introduction content of ZnO could be confirmed as 3 wt.%.

### 3.3. Electrochemical Tests

The ionic conductivity of the SPEs is an essential parameter assessing the electrochemical performance of the lithium batteries, which is applied to characterize the conductive ability of the SPEs for Li^+^ as the charge carrier. The value of the SPEs is generally less than that of the commercial liquid electrolyte, so improving the value of the SPEs via modification has become a research focus. The modification aims to reduce the crystallinity of the SPEs due to Li^+^ intercalation/deintercalation in the SPEs mainly occurring in the moving polymer chain segments distributed in the non-crystalline zones. A certain amount of ZnO (3 wt.%) contributed to enhancing the ionic conductivity because the crystallinity of the SPE was greatly reduced by nearly 71.31% when the content of ZnO was increased from 0 wt.% to 3 wt.%. The ionic conductivity of the SPEs could be determined by the following equation [13]:(2)$$\sigma =\frac{d}{S\times R_{b}}$$
where S refers to the area of the stainless-steel sheet, *d* signifies the thickness of the SPEs, and $R_{b}$ denotes the bulk impedance of the SPEs. 

S was controlled at a constant of 1.96 cm^2^, and d was accurately surveyed by a spiral micrometer (Table 3). $R_{b}$ was obtained by electrochemical impedance spectra (EIS) tests within the frequency scope of 100 kHz–0.01 Hz. Figure 4a showed the EIS of the SPEs in the symmetrical coin cells (SS (stainless-steel sheet)/SPE/SS) obtained at 60 °C, based on which the $R_{b}$ characterized by the intercept was measured (Table 3). According to Equation (2), σ of the SPEs was calculated as follows: 1.426 × 10^−4^ S·cm^−1^ (0 wt.% ZnO), 7.721 × 10^−4^ S·cm^−1^ (1 wt.% ZnO), 8.622 × 10^−4^ S·cm^−1^ (2 wt.% ZnO), 1.255 × 10^−3^ S·cm^−1^ (3 wt.% ZnO), 6.506 × 10^−4^ S·cm^−1^ (4 wt.% ZnO), and 5.239 × 10^−4^ S·cm^−1^ (5 wt.% ZnO). In general, the value of the SPEs with ZnO was higher than that without ZnO. However, with respect to the SPEs with ZnO, the value showed an inverted Λ-shaped trend, so the SPE acquired the highest value at 3 wt.% ZnO. The value was sharply improved by 779.87% when compared with that without ZnO. This result was also in agreement with the change in crystallinity when the content of ZnO changed. The values were also calculated at other temperatures (25 °C, 40 °C, and 80 °C) to study the influence of temperature on σ (Table 4). In this work, σ was gradually improved with the increase in temperature for a specific SPE, and it was first increased and then decreased with the increase in ZnO content for a specific temperature. 

The data pairs of lgσ and 1000/*T* were plotted in Figure 4b, demonstrating an approximately linear relationship expressed by the Arrhenius formula [20]:(3)$$\mathrm{lg}\sigma =-\frac{E_{a}}{2303R}\times \frac{1000}{T}$$
where *T* represents the absolute temperature, R represents the molar gas constant (8.314 J·mol^−1^·K^−1^), and $E_{a}$ represents the activation energy of Li^+^ transfer.

The calculated results indicated that $E_{a}$ was decreased from 46.26 kJ·mol^−1^ to 27.59 kJ·mol^−1^ with the increase in ZnO content from 0 wt.% to 3 wt.% and then improved to 34.12 kJ·mol^−1^ (4 wt.%) and 34.46 kJ·mol^−1^ (5 wt.%) upon introducing more ZnO. These observations indicated that Li^+^ could divert relatively easy in the SPE (3 wt.% ZnO).

In addition to the ion conductivity, the electrochemical window is another important indicator applied to appraise the stability of the SPEs, namely, a wide electrochemical window contributes to increasing the energy density of the batteries by choosing a cathode material with an upper potential without destroying the SPEs. LSV tests were performed on the SPEs with different ZnO contents under the voltage scope of 0–6.0 V and a certain scanning speed (5.0 mV·s^−1^) at 60 °C. The voltage referred to that between the working electrode (stainless-steel sheet) and the reference electrode (lithium sheet) in the asymmetrical coin cells (Li (lithium sheet)/SPE/SS). The curves of all SPEs in the voltage scope of 1–4 V were relatively smooth (Figure 4c), and the current was hardly changed. For the SPE without ZnO, the current was gradually increased when the voltage exceeded 4 V and was risen sharply when the voltage exceeded 5 V, implying that the reactions occurred in the SPE. When 1 wt.% and 2 wt.% ZnO was introduced into the SPEs, the platform could be prolonged up to 5.3 V, and the current showed a slight upward tendency as the voltage was further increased. With the increase in ZnO content to 3 wt.%, the platform was further increased to approximately 5.5 V, beyond which the current was also increased extremely slowly. When more ZnO was introduced, the electrochemical window presented the opposite trend, which was reduced to 5.0 (4 wt.%) and 4.5 V (5 wt.%). Thus, the suitable content of ZnO could greatly expand the electrochemical window. For the SPE without ZnO, the dehydrogenation reaction of PEO could occur when the external voltage exceeded the window (4 V), and the reaction became more intense as the external voltage increased [43]. In addition, the end group (–OH) of PEO was oxidized to the carboxyl group (–COOH). The active hydroxyl group and carboxyl group may react with the electrode to promote the degradation of the SPE [44]. ZnO is mainly synthesized at the end group of PEO, which as a cross-linking point changes the end groups of PEO, further enhancing the oxidation resistance of the SPE. The positive role in improving the electrochemical window resulting from the interaction between ZnO and PEO will be weakened when excessive ZnO causes agglomeration. Tan et al. [30] proposed a method for in situ synthesis of SiO_2_ particles in the SPEs. Their test results showed that the electrochemical window for Li/Li^+^ was extended to 5 V by the strong interaction among SiO_2_ particles, and PEO chains induced during in situ synthesis.

Besides the ionic conductivity and electrochemical window, the lithium-ion migration number ($t_{\mathrm{ion}}$) is regarded as another important parameter, which is used to characterize the percentage of the charge transfer of Li^+^ in the total charges. A high value is beneficial to the improvement in quick charge ability. $t_{\mathrm{ion}}$ of the SPEs can be quantitatively computed by the following computational equation [17]:(4)$$t_{\mathrm{ion}}=\frac{I_{s}\;\times \;\left(\Delta {V-I}_{0}R_{b}^{0}\right)}{I_{0\;}\times \;\left(\Delta {V-I}_{s}R_{b}^{s}\right)}$$
where ΔV is the polarization voltage applied to the coin cell; $I_{s}$, $I_{0}$, $R_{b}^{s}$, and $R_{b}^{0}$ are the steady-state current, initial current, bulk impedance post-polarization, and bulk impedance pre-polarization, respectively. 

The polarization (Figure 5(a0–a5)) and EIS curves plots (Figure 5(b0–b5)) of the SPEs in the symmetrical coin cells (Li (lithium sheet)/SPE/Li) were shown in Figure 5. The polarization curves presented the relationship between current and time at a constant potential window (ΔV = 10.0 mV). The polarization voltage (ΔV = 10.0 mV) referred to that between lithium sheet and lithium sheet in the symmetrical coin cells (Li (lithium sheet)/SPE/Li). $I_{0}$ and $I_{s}$ were obtained from Figure 5(a0–a5) (Table 5). The EIS curves of the coin cells before and after polarization were recorded (Figure 5(b0–b5)); a straight line with a slope of about 1 appeared in the low frequency area, and a semicircular curve appeared in the high frequency area. The bulk impedance ($R_{b}^{0}$ and $R_{b}^{s}$) of pre-polarization and post-polarization was acquired based on the intercept of the curves on the *x*-axis (Figure 5(b0–b5) (Table 5). Using the data from Figure 5, $t_{\mathrm{ion}}$ was calculated by Equation (4) (Table 5). The results presented a gradual uptrend as the ZnO content improved from 0 wt.% to 3 wt.% (0.645 for 0 wt.% ZnO, 0.810 for 1 wt.% ZnO, 0.835 for 2 wt.% ZnO, and 0.858 for 3 wt.% ZnO). However, the further increase in ZnO content played an opposite role due to the decrease in $t_{\mathrm{ion}}$ (0.788 for 4 wt.% ZnO and 0.771 for 5 wt.% ZnO). The SPE acquired the highest value at 3 wt.% ZnO, which showed a 33.02% improvement when compared with that without ZnO. As the physical cross-linking center, ZnO changed the structural state of the polymer chain segments and provided more paths for the transmission of Li^+^. In addition, the competition between ZnO and the anionic group of the lithium salt for binding to Li^+^ promoted the separation of the lithium salt and improved the transfer ability of Li^+^. However, superfluous incorporation of ZnO led to severe excessive cross-linking, which hindered the migration path of Li^+^ [34,39,40]. 

To assess the chemical stability of the SPEs and their long-term compatibility with metallic lithium, the SPEs without ZnO and with 3 wt.% ZnO and lithium metal plates were assembled into the coin cells with a sandwiched structure (Li (lithium sheet)/SPE/Li). The coin cells were subjected to constant current charge/discharge tests (1 h charge/1 h discharge as a cycle) with the applied current of 20.0 μA·cm^−2^ at 60 °C. The voltage referred to that between lithium sheet and lithium sheet in the symmetrical coin cells (Li (lithium sheet)/SPE/Li). As shown in Figure 6a–c, for the coin cell without ZnO, the voltage piecemeal was reduced from 0.0936 V to 0.0827 V within the first 160 h, which was mainly caused by the generation of a thin solid electrolyte interphase (SEI) film. The thin SEI film as the medium could improve the affinity between the SPE and electrode, which promoted the transportation of Li^+^ and reduced the overpotential during the charge/discharge processes. However, as the charge and discharge proceeded, the film gradually became thicker and rougher due to the uneven deposition of Li^+^, casing the internal resistance of the cell to increase (the voltage rose to 0.1311 V in 549 h) [21,45]. Figure 6b showed that the voltage value mutated into 0 V after 549 h, indicating that the coin cell was completely ineffective. This result may be attributed to the overgrowth of lithium dendrites piercing the electrolyte and causing the short circuit inside the coin cell [5,11,17]. In comparison, the voltage of the coin cell containing 3 wt.% ZnO was first decreased from 0.0883 V to 0.0787 V within 55 h and then maintained a comparatively stable value of approximately 0.0781 V up to 1940 h (Figure 6c). ZnO can interact with Li metal to a certain extent and reduces the reduction onset potential of the electrolyte, resulting in the formation of a stable SEI film. A stable SEI film not only effectively reduces the interface resistance, but also inhibits the growth of lithium dendrites. This phenomenon had also been confirmed in the other related reports [46,47]. The positive influence in mechanical property of the SPEs with ZnO was responsible for the significant difference in stability between the two. During long-term charging/discharging, the lithium dendrites nucleated, grew on the lithium plate inevitably, and pierced through the SPEs, causing the short circuit through direct contact of the two lithium sheets. Therefore, the mechanical property of the SPEs played an essential role in stopping and retarding the overgrowth of lithium dendrites. As shown in Table 2, the mechanical properties of the SPEs in terms of strength and toughness were greatly improved by introducing ZnO into the SPEs, which endowed the SPEs with strong resistance to plasticity and cracking when the growing lithium dendrites came into contact with the SPEs. Consequently, the pierced difficulty and duration of the SPEs were greatly delayed because the growth of lithium dendrites was obviously inhibited. This speculation can be further proved by observing lithium dendrites’ morphologies after the coin cells (0 wt.% ZnO and 3 wt.% ZnO) were subjected to long-term charge/discharge tests (Figure 6(d1–d4)). As shown in Figure 6(d1), the electrode surface in the coin cell containing 0 wt.% ZnO was quite rough, to which a large number of fluffy clusters with a honeycomb structure adhered. The high-magnification image (Figure 6(d2)) showed that those clusters were composed of many flower-like lithium dendrites with very sharp edges, among which there was a huge difference in height. Thus, the lithium dendrites could grow independently and rapidly in the horizontal and vertical directions. The lithium dendrites in some areas may grow preferentially along the direction perpendicular to the electrode surface, coming into direct contact with the SPE and piercing through it due to sharp protrusions, which could cause the short circuit of the cell under relatively short cycling (549 h). When 3 wt.% ZnO was introduced, the electrode surface became smooth and dense (Figure 6(d3)). The high-magnification image (Figure 6(d4)) showed that many irregular spherical particles with smooth edges piled up tightly, among which there was no significant difference in height. These results demonstrated that the growth of lithium dendrites was strongly inhibited along the direction perpendicular to the electrode surface, which was attributed to the great improvement in resistance to deformation and cracking of the SPE with 3 wt.% ZnO. Therefore, the lithium dendrites grew along the horizontal direction, resulting in the formation of the smooth and dense dendrite layer. Correspondingly, the cycling duration was significantly prolonged (no short circuit observed up to 1940 h). In order to further identify the difference between the SPEs with in situ synthesized ZnO and directly-added ZnO, the SEM result of the lithium sheet disassembled from the coin cell containing the SPE with directly-added 2 wt.% ZnO subject to the charge/discharge tests was used for comparison. Seeing Figure 12c–d illustrated in Reference [26], a small number of dendrites were sporadically distributed on the surface, which was smoother than that (without ZnO), but rougher than that with (3 wt.% in situ ZnO). This means that the introduction of ZnO can effectively retard the overgrowth of the lithium dendrites and promote their uniform growth on the whole surface. Comparatively speaking, the in situ synthesized ZnO can endow the SPE with better mechanical properties than directly-added ZnO, resulting in the further-improved ability to inhibit the overgrowth of lithium dendrites.

The LFP (cathode)/SPE-0 wt.% ZnO/Li (anode) and LFP/SPE-3 wt.% ZnO/Li coin cells were used to investigate the rate capacity at 60 °C. In the case of a cutoff voltage of 2.7–4.2 V (between LFP and Li), the charge/discharge cycle tests of the coin cells were subjected at six different current densities (0.1 C, 0.2 C, 0.5 C, 1 C, 2 C, and 3 C) with three cycles at each current density in turn and then 50 cycles at 1 C. For the coin cell assembled by the SPE without ZnO, the results were shown in Figure 7(a1–a3). The first discharge specific capacities of the coin cells under the first three current densities were 141.5 (0.1 C), 136.7 (0.2 C), and 107.2 mAh·g^−1^ (0.5 C), and their coulombic efficiencies were 94.65%, 103.80%, and 79.23%, respectively. However, the charge/discharge capacities at 1 C and 3 C in Figure 7(a2) were also close to zero, so the charge/discharge curves were almost invisible in Figure 7(a1). Thus, the lithium dendrites in the coin cells sufficiently grew when subjected to three cycles at 0.1 C, 0.2 C, and 0.5 C, and they may subsequently pierce the SPE and cause a short circuit when applying a large current density (1 C and 3 C) [28,34]. For the coin cell with the SPE containing 3 wt.% ZnO (Figure 7(b1–b3)), its first discharge specific capacities at 0.1 C, 0.2 C, 0.5 C, 1 C, 2 C, and 3 C were 155.4, 149.7, 126.7, 109.3, 82.1, and 58.3 mAh·g^−1^, respectively, and their coulombic efficiencies were 98.35%, 99.01%, 99.84 %, 99.91%, 99.03%, and 97.65%, respectively. No short circuit phenomenon was observed. When the coin cells underwent 50 cycles at 1 C, their discharge specific capacity and coulombic efficiency reached 106.8 mAh·g^−1^ and 99.91%, respectively, and these indexes were very stable during the whole cycle. Compared with the first discharge specific capacity of 0.1 C, a high capacity retention rate of approximately 68.73% was retained after 50 cycles at 1 C. The results clearly displayed that the introduction of ZnO into the SPEs could tremendously improve the rate capacity, which was also closely connected with the enhancement in mechanical property.

The electrochemical performance of the SPE with 3 wt.% ZnO was compared with that of our previous work and the other related works (Table 6). In our previous study, six contents of ZnO were directly added into the SPEs (0–5 wt.%), and their effects on the electrochemical performance of the SPEs were investigated. The results proved that the SPE with 2 wt.% directly-added ZnO demonstrated the most outstanding performance among the tested SPEs (Table 6). The SPE with in situ 3 wt.% ZnO presented better electrochemical performance than that with directly-added 2 wt.% ZnO in terms of ion conductivity, migration number, electrochemical window, cycle performance, and rate performance. Moreover, the electrochemical performance of the SPE with in situ 3 wt.% ZnO was generally superior to that reported in the other references (Table 6).

## 4. Conclusions

(1)PEO–COO^−^-modified ZnO was successfully prepared by the three-step method. The SPEs were fabricated by blending PEO, PVDF, LiClO_4_, and in situ synthesized ZnO (0–5 wt.%) introduced with PEO-COO^-^-modified ZnO as the intermediate.(2)The crystallinity of the SPEs reached the lowest value (2.04%) when adding a suitable content of in situ ZnO (3 wt.%). The SPE with 3 wt.% in situ ZnO also presented the best comprehensive mechanical properties in terms of the highest strength (2.736 MPa in yield strength, 4.749 MPa in tensile strength, and 2.855 MPa in breaking strength), plasticity (442.599% in elongation at break), and toughness.(3)The SPE with 3 wt.% in situ ZnO presented the best electrochemical performance, including the highest ion conductivity (1.378 × 10^−4^ S·cm^−1^ at 25 °C and 1.255 × 10^−3^ S·cm^−1^ at 60 °C) and migration number (0.858 at 60 °C), the widest electrochemical window (5.5 V at 60 °C), and the most outstanding rating capability (a capacity retention of 68.73%) and cycle stability (more than 1940 h).(4)Compared with directly-added ZnO, the in situ ZnO further improved the electrochemical performance of the SPEs.

## Data Availability

Not applicable.

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
