# Peer review of "A Strategy for Preparing Solid Polymer Electrolytes Containing In Situ Synthesized ZnO Nanoparticles with Excellent Electrochemical Performance"

_nanomaterials, 2022, doi:10.3390/nano12152680_

Round 1

Reviewer 1 Report

I found the paper entitled "A strategy for preparing solid polymer electrolyte containing in-situ synthesized ZnO nanoparticle with excellent electrochemical performance" very accurate and suitable for the publication in Nanomaterials. The polymer electrolytes are well characterized and the description procedures are clear. 

I suggest the following correction:

1) In the "2.2.3 Preparation of cathode" paragraph, the acronym SP has not been introduced before. It is referred to the acetylene black; however, it should be clarified better. 

2) In Table 1 a line with the crystallinity of each polymer should be added, for the sake of clarity.

3) I found the manuscript too long. I suggest the authors to put some minor results and table in the supporting information.  

Author Response

         Firstly, thanks for your careful review for my paper. Based on your advice, we had revised my paper in detail, and the revised parts had been clearly marked in red in revised manuscript [Changes highlighted]. Please see the attachment.

Reviewer 2 Report

In this papers authors describe the optimization of content of ZnO in solid polymer electrolytes that were synthesized via insitu process to achieve high iconic conductivity and good mechanical properties. The authors have analyzed the chemical and crystalline structure of the composite material using various techniques. Subsequently they have used various approaches to study the the effect of ZnO content on the composites’ crystallinity, overall  mechanical and electrochemical properties (Li/Li symmetric cell and full cells using LFP). The results shown, illustrate that 3% ZnO containing polymer electrolyte has the best mechanical and electrochemical performance. The authors have attributed this to the good interaction between ZnO and the polymer material and its effect on the overall mechanical properties of the composite. This is an interesting study, however, the paper lacks several key baseline results and several conclusions made are not supported by the presented data. Also the general language of the manuscript requires major revisions to make the manuscript clear and easy to read.

The authors should address some of the following issues:

1.       The figure captions and their descriptions within the text are very confusing.

2.       The authors have highlighted in-situ ZnO synthesis process within the manuscript but failed to compare this synthesis process to simple addition of ZnO to the polymer matrix in major part of the manuscript (they only added a table at the end comparing the electrochemical results). For a better discussion it would have been better to add another sample just with 3 or 2wt% ZnO directly added to the matrix from the start of the manuscript.

3.       The authors attribute the good electrochemical properties of the 3wt% ZnO containing composite to the effect of ZnO on the overall mechanical properties of the composite. Specially they highlight this when explaining the symmetric cell data. However they fail to explain how ZnO effects the SEI layer between the electrolyte and Li metal.

4.       In the introduction, the authors fail to explain why ZnO was chosen over other metal oxides for this study.

5.       There is no clear evidence presented for integration of ZnO nanoparticles with polymer matrix. There is no results stating the size of ZnO particles and there is no clear evidence of ZnO agglomeration above 3wt% ZnO content. There is a need for better imaging (SEM or TEM).

6.       The authors talk about the effect of ZnO on the roughness of samples illustrated using SEM imaging. It would have been better to acquire simple roughness measurements since SEM images of the 3wt% ZnO sample show some roughness compared to the 2wt% sample.

7.       The authors show post-mortem SEM images of the Li metal surface after cycling. The cell disassembly should be covered in the experimental methods. Also the authors should describe the adhesion of lithium metal to the polymer electrolyte in more detail.

Author Response

          Firstly, thanks for your careful review for my paper. Based on your advice, we had revised my paper in detail, and the revised parts had been clearly marked in red in revised manuscript [Changes highlighted].Please see the attachment.

Reviewer 3 Report

 in this article, authors report in situ synthesis of ZnO nanoparticles embedded into PEO based solid polymer electrolyte. Later, they applied this electrolyte as SPE in Li metal battery. The synthesis, general and specific characterization methods, final device demonstration are well reported. Moreover, structure-property-performance relationship was well designed and demonstrated. Based on this, I suggest to accept this article, after addressing minor issues.

1.       English language and grammar should be improved and typos should be corrected in many places.

               For instance, first sentence of abstract, line 34 (page 1), etc.

2.       Whenever voltage values are reported, authors should be referred against the reference electrode, at least in the beginning of each section.

3.       I encourage authors to complement SEM studies also by using AFM.

4.       I suggest to use elongation at break instead of Elongation after breaking in the context of mechanical properties analysis.

5.       Nyquist plots should be given in the symmetric scale.

Author Response

(The authors gave the same response as above.)
